# Breath Insights: Advancing Lung Cancer Early-Stage Detection Through AI Algorithms in Non-Invasive VOC Profiling Trials

**DOI:** 10.3390/cancers17101685

**Published:** 2025-05-16

**Authors:** Bernardo S. Raimundo, Pedro M. Leitão, Manuel Vinhas, Maria V. Pires, Laura B. Quintas, Catarina Carvalheiro, Rita Barata, Joana Ip, Ricardo Coelho, Sofia Granadeiro, Tânia S. Simões, João Gonçalves, Renato Baião, Carla Rocha, Sandra Alves, Paulo Fidalgo, Alípio Araújo, Cláudia Matos, Susana Simões, Paula Alves, Patrícia Garrido, Marcos Pantarotto, Luís Carreiro, Rogério Matos, Cristina Bárbara, Jorge Cruz, Nuno Gil, Fernando Luis-Ferreira, Pedro D. Vaz

**Affiliations:** 1Unidade de Pulmão, Centro Clínico Champalimaud, Fundação Champalimaud, 1400-038 Lisboa, Portugal; bernardo.raimundo@research.fchampalimaud.org (B.S.R.); maria.pires@research.fchampalimaud.org (M.V.P.); laurabquintas@gmail.com (L.B.Q.); catarina.carvalheiro@fundacaochampalimaud.pt (C.C.); rita.barata@fundacaochampalimaud.pt (R.B.); ricardo.coelho@fundacaochampalimaud.pt (R.C.); ana.granadeiro@fundacaochampalimaud.pt (S.G.); taniasim@hotmail.com (T.S.S.); joao.goncalves@fundacaochampalimaud.pt (J.G.); renato.baiao@fundacaochampalimaud.pt (R.B.); carla.rocha@fundacaochampalimaud.pt (C.R.); claudia.matos@fundacaochampalimaud.pt (C.M.); susana.simoes@fundacaochampalimaud.pt (S.S.); patricia.garrido@fundacaochampalimaud.pt (P.G.); marcos.pantarotto@fundacaochampalimaud.pt (M.P.); luis.carreiro@fundacaochampalimaud.pt (L.C.); rogerio.matos@fundacaochampalimaud.pt (R.M.); jorge.cruz@fundacaochampalimaud.pt (J.C.); nuno.gil@fundacaochampalimaud.pt (N.G.); 2Departamento de Engenharia Electrotécnica e de Computadores, Faculdade de Ciências e Tecnologia, Universidade Nova de Lisboa, Quinta da Torre, 2829-516 Caparica, Portugal; pm.leitao@campus.fct.unl.pt (P.M.L.); mfp.vinhas@campus.fct.unl.pt (M.V.); 3Serviço de Radiologia, Centro Clínico Champalimaud, Fundação Champalimaud, 1400-038 Lisboa, Portugal; joana.ip@fundacaochampalimaud.pt; 4Unidade de Ensaios Clínicos, Centro Clínico Champalimaud, Fundação Champalimaud, 1400-038 Lisboa, Portugal; sandra.alves@fundacaochampalimaud.pt; 5Unidade de Risco e Diagnóstico Precoce, Centro Clínico Champalimaud, Fundação Champalimaud, 1400-038 Lisboa, Portugal; paulo.fidalgo@fundacaochampalimaud.pt (P.F.); alipio.araujo@fundacaochampalimaud.pt (A.A.); 6Serviço de Pneumologia, Centro Hospitalar e Universitário Lisboa Norte, 1649-035 Lisboa, Portugal; alvespaula57@gmail.com (P.A.); cristina.barbara@chln.min-saude.pt (C.B.)

**Keywords:** lung cancer, early detection, breath analysis, volatile organic compounds, artificial intelligence, non-small cell lung cancer

## Abstract

Lung cancer (LC) stands as the leading cause of cancer-related deaths. Fostering innovative screening methods for early detection could improve survival rates. Volatile organic compounds (VOCs) found in exhaled breath air are becoming a relevant opportunity for early cancer detection, including lung cancer, without invasive procedures or high costs. Unlike traditional approaches that target specific compounds, this study analyzes overall compositional profiles, maximizing detection efficiency. The findings suggest that breath-based screening, combined with artificial intelligence (AI), could be a reliable and non-invasive method to detect LC earlier, improving prognosis and making diagnosis easier. This approach could help reduce the challenges researchers have faced in earlier LC detection efforts. The results highlight the potential of AI-driven techniques in revolutionizing early cancer detection for clinical use.

## 1. Introduction 

Lung cancer (LC) is the leading cause of cancer morbidity and mortality worldwide, with almost 2.5 million new cases and over 1.8 million deaths in 2022. It is responsible for 12.4% of the cancers diagnosed globally and approximately one-fifth (18.7%) of cancer deaths, ranking first among men and second among women for both incidence and mortality, with a male-to-female lung cancer incidence and mortality ratio of around two [1]. Despite advances in various treatment modalities, the long-term survival rate remains low. Although LC at stage I reaches a five-year survival rate up to 80%, only 25% of patients are diagnosed at this early stage [2,3]. An early detection of LC would change the disease prognosis, but there is a lack of active implementation of screening programs around the world that could change the LC landscape [4,5,6].

The National Lung Screening Trial (NLST), a U.S. multicenter, randomized controlled trial that enrolled more than 53,000 people, reported a significant 20% improvement (*p* = 0.004) in LC mortality and a 6.7% improvement in overall mortality in individuals undergoing low-dose computed tomography (LD-CT) compared with those undergoing thorax X-ray [7]. Seventy percent of detected LC cases were in early stage [8].

Since the publication of the NLST in 2011 [7], several other trials have evaluated the impact of LC screening. The Dutch–Belgian Randomized Lung Cancer Screening Trial (NELSON) randomized high-risk individuals to annual LD-CT versus observation [9]. The trial involved more than 15,000 people aged 50–75 years with a high tobacco intake, and screening was performed at 0, 1, 3, and 5.5 years. Approximately 84% were male individuals, and for those who were randomized for the LD-CT arm, after 10 years of follow-up, LC-related death reduction was 24%. In women, the reduction in LC-specific mortality was higher [3]. Among screened male participants, up to 68% of lung cancers were at stages I and II.

The consistent findings evidenced that an annual LD-CT detects LC at earlier stages and reduces LC-related mortality among high-risk individuals. Unfortunately, there are many issues contributing to the low adhesion to LC screening programs, including a lack of patient and clinician awareness of the mortality benefit of LD-CT, clinician concerns about radiation exposure, false-positive results, overdiagnosis and overtreatment, health system resource utilization, and cost-effectiveness [3,7,8,10,11].

Recently, relevant clinical trials established the analysis of volatile organic compounds (VOCs) in exhaled breath as a promising non-invasive method for cancer screening [12,13,14,15,16,17,18]. The results arising from those studies, although aiming towards similar objectives, relied on different chemical analysis techniques, such as mass spectrometry (the vast majority) under different available approaches, ion-mobility spectrometry or e-nose systems, all of which have their virtues and limitations. Exhaled breath contains a complex mixture of volatile and non-volatile organic compounds produced as end-products of metabolism. The VOCs resulting from metabolic activity of cancer cells and tumor microenvironment are eliminated into the bloodstream and ultimately excreted by diffusion across the pulmonary alveolar membrane and exhaled through breath, leading to a unique breath signature.

The European Respiratory Society issued a technical standard addressing challenges for such an approach [19]. Nevertheless, due to a lack of consensus across most of published reports and projects [13], VOC methods (collectively named “breathomics”) and their use as tools are still in their early stages [20,21].

This study aimed to assess the possibility of using VOC analysis in exhaled breath as biomarkers for LC screening. It intended to demonstrate enhanced sensitivity and specificity for the correct classification of lung cancer detection by combining AI tools with a qualitative VOC profile test from exhaled breath (Figure 1) [22,23,24]. The profiles were determined by gas chromatography with ion-mobility spectrometry (GC-IMS), which provided a highly efficient separation of the VOC components while disregarding the qualitative nature of the VOCs. IMS is a mass spectrometry technique that can be used to separate complex mixtures of molecules based on their different mobility (which relate to size and mass). It has several advantages over traditional mass spectrometry as it operates at room temperature, ambient pressure, does not require specific gases, and does not suffer from sensor drift while maintaining the same level of sensitivity.

The VOC profiles were analyzed using machine learning (ML) algorithms, generally called “artificial intelligence” (AI), leading to the development of models, able to generate predictions or conclusions for new proposed cases based on a training set, turning AI into a departing requirement to find reasons for a probabilistic outcome over a patient sample [23,24]. 

## 2. Methods

### 2.1. Volunteer Recruitment and Characterization

We conducted a multicentric, observational study from July 2020 to December 2023 at two different clinical centers in Lisbon: Champalimaud Clinical Center—Champalimaud Foundation and Hospital Pulido Valente–Hospital Lisboa Norte (part of Centro Hospitalar Universitário Lisboa Norte—CHULN). 

The study protocol and all amendments were approved by the ethics councils of the respective clinical centers, ensuring that the study was conducted according to Good Clinical Practice standards. All volunteers were briefed and provided their written informed consent before enrollment. Further details can be found in the Appendix A in this article.

The volunteers’ background data—demographic profiles, cigarette smoking history—were collected during the recruitment appointment. For the LC group staging, pathological findings were also recorded. LC staging followed the 8th edition of the American Joint Committee on Cancer (AJCC) TNM staging system [25]. Samples and data from LC patients were collected before undergoing any treatment, including surgery.

### 2.2. Exhaled Breath Sampling

Breath samples were collected at the clinical site where the patient was recruited. When sampling exhaled breath air, all enrolled volunteers from both groups fasted (food and any drinks apart from water) for at least four hours before sampling to avoid possible effects of food or its metabolites on the profile of volatile compounds. All volunteers were also required not to smoke or have their oral hygiene for at least two hours before breath sampling. Atmospheric air from both the sample collection rooms and the sample collection device was also analyzed to investigate the effects of background VOCs on collected breath samples.

Breath sampling was made possible using the ReCIVA breath sample system (Owlstone Medical, Cambridge, UK), which allowed the collection of exhaled breath air under reproducible conditions. Clean air was constantly supplied to the volunteer’s mask through a pump connected to an active charcoal scrubber (CASPER system; Owlstone Medical, Cambridge, UK) at a flow rate of 500 mL/min. Collection of breath was performed directly with a mask holding four thermal desorption (TD) tubes (Bio-monitoring, inert coated tubes, Markes International Ltd., Bridgend, UK). TD tubes were made of passivated stainless steel (denoted inert as per the manufacturer’s specification) with a mixture of two porous retention materials: Tenax TA (based on 2,6-diphenyl-p-phenylene oxide polymer) and Carbograph 5TD (a non-specific carbon sorbent), allowing for retention of C_4_-C_32_ VOC analytes. Breath sampling was non-invasive, with volunteers performing normal tidal volumes using the ReCIVA system, which was set to collect 2 L of full exhaled breath that was transferred into the four replicate TD tubes. 

### 2.3. GC-IMS Analysis

Analysis of all samples was carried out using a commercially available GC-IMS instrument (Lonestar, Owlstone Medical, Cambridge, UK). In this case, the separation step was performed in a Trace 1310 GC gas chromatograph graph from Thermo (Thermo Fisher Scientific, Waltham, MA, USA), equipped with an HP-PLOT U column, 30 m length, 0.32 mm internal diameter, and 10.0 µm film thickness, coupled to an IMS detector (Lonestar FAIMS Analyzer, Owlstone Medical, Cambridge, UK). The initial temperature of the GC program was 40 °C (held for 2 min), followed by a heating ramp of 10 °C/min until reaching 130 °C, maintained isothermally until a total analysis time of 30 min was completed. The Lonestar IMS instrument uses field-asymmetric ion-mobility spectrometry (FAIMS) for the detection of analytes. It relies on an oscillating electric field to separate different gas-phase ions based on their different mobility across an electrical field relating to size and mass. The FAIMS detector was operating with the dispersion field (DF) kept constant at 45% and the compensation voltage (CV) between -6 V to +6 V.

The resulting data reveal a profile of a “chemical fingerprint” (the total VOC chemical composition) of a given sample. Each volunteer provided a set of four replicate profiles arising from the four tubes.

### 2.4. Classification Methods

The analysis of exhaled breath air with a spectrometer produced VOC profiles with a numeric value for the intensity of each compound. The spectrometer recorded the intensity of each VOC found in the sample for each patient, which could be systematized in a plot [22]. Thereafter, the numeric array obtained for each patient was classified according to its corresponding health status as evaluated in current clinical practice. 

In the present study, the dataset was made of spectrometry-generated data resulting from different exhaled air samples used to train selected ML algorithms. The process consisted of systematically training the AI algorithms with samples from both healthy individuals and LC patients. Supervised learning algorithms, including support vector machines (SVM), multilayer perceptron (MLP) and K-nearest neighbor (KNN), were applied to the classified dataset. Once trained, the algorithms were evaluated with key performance indicators on their performance and accuracy in classifying new cases. The model was trained using the 10-fold cross-validation method. Further details are provided in the Appendix A.

## 3. Results

### 3.1. Patient Characterization

The study recruited 196 validated participants, 73 of whom were in the lung cancer (LC) group and 123 in the healthy control group (HC), hereafter denoted as the control group (C). All the volunteers in the LC group had their disease histologically proven by biopsy, and those in the control group could not have any known cancer diagnosis between enrollment and sample collection. The background demographic data for all the participants in this study are shown in Table 1.

Smoking status was balanced across both groups (31.5% in LC and 31.7% in C), while a more significant proportion of never smokers in the C group (54.5% vs. 30.1%) was found. Current and former smokers represented 69.9% of the LC population versus 43.7% in the healthy volunteers (C group). 

The most frequent histology for LC was adenocarcinoma (*n* = 49, 67.1%) followed by squamous cell carcinoma (*n* = 6, 8.2%) and small cell carcinoma (*n* = 5, 6.9%). More than half (57.5%) of all LC patients presented stage 0, IA, or IB. For a complete description of the histology found for the patients enrolled, please refer to the Appendix A. No adverse events were registered due to the exhaled breath collection procedure.

### 3.2. Chemical Analysis of Exhaled Breath Samples

Exhaled breath is a body fluid displaying a rich yet complex composition of volatile and non-volatile metabolites and exhaled volatile organic compounds (VOCs) arising from the body’s metabolic activity, which mirror the health condition at a given moment in time.

In this study, the use of GC-IMS did not allow for the identification of individual component metabolites. Instead, the complete chromatographic profile of the intensities of all detected analytes obtained for each volunteer was used for classification according to their status (C or LC groups). Figure 2 displays typical profiles for both the C and LC (at early and advanced stages of the disease) groups. There were significant differences between groups, which allowed for establishing a typical VOC profile for each group.

As shown in Figure 2, the profiles from both groups differed in their overall shape and total integrated intensity (area under the curve, AUC), with AUCs higher for advanced-stage patients (Figure 2B). The profile of the healthy individuals displayed the lowest AUC of the set (Figure 2C). Figure 2D shows a superposition of all three profiles, stressing the differences among them. It becomes clear that the profiles differ in their chemical composition as given by both signals appearing at different retention times (different analytes) and different peak intensities (different relative concentrations). This demonstrates that the conditions associated with the chromatograms shown in Figure 2D differ in their chemical composition; additionally, similar analytes can be assumed to be present at different concentrations.

### 3.3. Data Processing and Machine Learning

We hypothesized that the difference observed in the exhaled breath profiles could open the way for classification algorithms to be used [22]. We used a dataset of samples from healthy individuals (C, 123 individuals, 492 samples) and patients with diagnosed lung cancer (LC, 73 individuals, 292 samples), totaling 784 observations. ML methods were applied to classify the samples, using supervised models to classify the datasets: decision tree (DT), random forest (RF), logistic regression (LR), support vector machine (SVM), multilayer perceptron artificial neural network (MLP), and K-nearest neighbor (KNN).

Model training also included distinguishing between healthy controls from lung cancer at stage IA and among the LC group to distinguish different histology (Table 2). We aimed to assess this tool’s sensitivity for screening early-stage lung cancer while also assessing whether adenocarcinomas could be detected independently of any other histology types.

According to Table 2, all models showed high specificity and NPV marks for LC stage IA and all-stages LC detection, with inferior performance for histological classification.

The ROC curves (Figure 3) for the best-performing classification methods displayed high AUC values for both “*C* vs. *LC all stages*” and “*C* vs. *LC stage IA*” classification. Furthermore, narrowing the classification between healthy controls and patients restricted to stage IA did not induce any changes in ROC curves, as confirmed by the AUC values (Figure 3).

The AUC values obtained along with the confidence intervals (Table 3) confirm the values for this metric displayed on the ROC curves.

Furthermore, DeLong’s test (Table 4) for the comparison between the methods shows that for every case, the given *p*-values are uniformly higher than 0.05. In statistical terms, this means there is no significant difference, suggesting performance differences may be due to random variation. Although KNN and SVM had a slightly higher value, that may not be the case for another iteration. 

## 4. Discussion

Early detection and diagnostics (ED&D) of LC is the matter of many efforts. The current recommendation is to perform annual an LD-CT scan in high-risk patients, which has been proven to reduce the risk of death from LC. However, the high cost, limited availability, false-positive results, overdiagnosis, and overtreatment are some of the obstacles to carrying out screening programs based on this method [10,26,27,28].

Weighing all these facts, there is the need for complementary screening methods, simpler, less expensive, and easily accessible to the population, allowing faster clinical decisions and ultimately improving disease outcomes. Recent systematic reviews covering several studies addressed the use of exhaled breath analysis for the detection of cancer [12,13,20,21,29,30]. The review by Hannah et al. included sixty-three studies and demonstrated that extensive incoherence existed among them, reasons being the lack of standardization and quality control [13]. Recent reviews addressed what is missing to bring analysis of exhaled breath into the clinic. They concluded that, despite some caveats, it can be achieved if standardization procedures can be implemented, ranging from breath sample collection and analysis through instrumentation up to reporting and validation of results [21,30]. 

The present study shows that analysis of profiles of exhaled breath alongside standardization of procedures and quality control can be an effective and feasible method for LC detection. In addition to the non-invasive nature and lack of radiation exposure, this methodology could be less expensive than the currently recommended available method. Breath collection and analysis can be implemented at points of care and easily accessible to the population, making it an attractive method in LC screening and, conceptually, it can offer an alternative tool in addition to present options [19,31].

### 4.1. Using Exhaled Breath Profiles for Lung Cancer Screening

In this study, GC-IMS was used as an efficient, sensitive, and powerful technique for analysis of exhaled breath [32]. Gas chromatography (GC) separates complex mixtures of VOCs into their individual components. IMS provides a second dimension of separation based on the interaction of an electric field with ions of the individual’s VOCs while also detecting them. This technique has much fewer limitations to be implemented in clinical practice than the GC-MS counterparts. For instance, the latter is usually more expensive, requiring more demanding installation infrastructure and more highly trained, highly skilled people to operate it. In contrast, the former is usually far simpler to set up and operate, making it easier to be deployed at clinical sites and may also be portable. Another advantage of IMS is that the ionization is very mild and does not yield any fragment ions, i.e., only the molecular ions are observed. This is a major advantage over mass spectrometry, where fragmentation is always present despite mild ionization methods that are also usually available. As a drawback, GC-IMS lacks the same power of identifying a specific compound or a possible chemical structure, although the detection capability (sensitivity) outperforms that of GC-MS. GC-IMS yields complete yet complex 2D separation chromatograms that offer ample information from a sample. The wealth of information provided in the 2D matrices can be used to classify exhaled breath profiles for LC detection, as we have recently demonstrated [33]. However, the approach presented in this study used a simplified version of those 2D profiles by projecting the matrices into normal 1D plots (Figure 2), resulting in better management of the datasets without compromising performance of the classification based on AI algorithms (Table 2). This methodology of pattern recognition using AI-based classification algorithms allowed recognition of specific patterns in the VOCs related to LC without identification of the specific VOC components.

If analysis of exhaled breath mirrors metabolic activity, its interpretation is a complicated task. The main issue relates to the fact that sources of exhaled VOCs are not yet well understood. Although several studies tried to address this, they lacked coherence among them [13,19,29]. Oxidative stress is recognizably the main VOC production mechanism, arising not only from cell metabolism but also from microbiota or even exogenous sources, thus explaining VOC qualitative and quantitative variations [34,35,36,37]. In those individuals with LC, VOCs can originate additionally from within the tumor or its microenvironment [38,39,40].

As such, in a scenario of LC after its onset, the set of VOCs will differ from the previous condition without the disease [41,42]. By weighing all this, one can associate the VOC profiles (disregarding any qualitative information) to a specific condition and defining, therefore, the overall VOC profile as the biomarker for LC detection. This makes the methodology more pragmatic and less prone to failure [33,43]. Gordon et al. reported a similar approach in 1985, relying on the profile of GC-MS data [44]. In that work, the authors stated that discriminating statistically between healthy and lung cancer profiles would not require knowledge of the chemical identity of the peaks [44].

In our study, the classification of VOC profiles achieved high accuracy across all three AI algorithms for the “*C* vs. *LC all stages*” with the best performing algorithm being KNN (87% sensitivity, 92% specificity) with the others (SVM and MLP) coming close (Table 2). Given that the NPV scores were very high (Table 2), it may indicate that this test is reliable at ruling out LC, enabling clinicians to be assisted by a consistent tool for LC detection. Since these are preliminary results, a larger study group aiming at validation is required. Notwithstanding this limitation, these results support the potential use of this method for LC screening. This is even more relevant given the ability to detect LC at early stages as per the results of the “*C* vs. *LC stage IA*” classification with an accuracy of 90% and the NPV reaching 96% using the KNN algorithm (Table 2).

Attempts at identifying LC histology were also pursued, showing the feasibility of the analysis of VOC profiles to successfully achieve it (Table 2). The quality and performance indicators were lower with values ranging between 70 and 80%, most probably due to the smaller size of the dataset, which took into consideration only the LC group. Yet this demonstrated the feasibility of the method as both a screening and stripping tool.

### 4.2. Study Population and Breath Test Standardization

All recruited individuals across both groups had noticeable differences at several levels. For instance, the LC group was on average older than the C group, although both age ranges overlapped. When comparing the ages of individuals in the healthy group to those diagnosed with lung cancer, it is almost inevitable that the lung cancer group will exhibit a higher median age. This is not necessarily indicative of a fundamental difference in the underlying susceptibility to the disease across all ages. Instead, it largely reflects the typical pattern of LC diagnosis. LC is often a disease that develops over many years, and symptoms may not become apparent until later stages. Consequently, diagnosis frequently occurs in older individuals who have had more time for the disease to progress to a detectable stage. Therefore, the higher median age in the lung cancer group is significantly influenced by this delay in diagnosis. It does not automatically imply that younger individuals are somehow immune, or that the risk of developing LC is only significant at older ages. It simply reflects the reality that the disease is often identified later in life [45]. In essence, the higher median age in the LC group is more a consequence of the disease’s progression and detection timeline than a direct measure of when the biological processes leading to lung cancer are most active or prevalent. Yet, no statistical significance was found when classifying the exhaled breath data according to age, indicating that the methodology is not biased by age. Smoking status was found to have differences. The fraction of current smokers was similar, but both groups differed in the remaining subcohorts (Table 1).

The lack of standardization and quality control was claimed as the probable reason for the inconsistent results found in many previous studies, as discussed elsewhere [13,46]. In this study, we selected a strict population based on a single disease and applied standardization procedures for patient preparation and breath sample collection, among others (see Methods) [13,19,20,46].

In this way, all volunteers enrolled followed the same preparation protocol prior to sample collection: fasting (at least 4 h), no smoking (at least 2 h), no medication (at least 6 h), no oral hygiene (at least 2 h); no creams, perfume, or lotions on the face. With these criteria for the collection protocol, we introduced a standardization procedure that made the exhaled breath air sample collection similar to any other clinical test, such as a blood test or a radiology imaging exam. In addition, when analyzing the exhaled breath air samples, the GC-IMS method measured and detected only the very volatile fraction of the (bio)chemical components from those samples. This avoided the presence in the analysis of known analytes commonly associated with smoking. The hospital environment was also ruled out as a confounder. This was confirmed by a two-fold procedure. The first was by sampling environmental air samples, whose patterns were as intense as the patterns found for the ReCIVA collection device itself, meaning that when compared with the actual samples from patients, the intensity of those profiles were at the background level. The second procedure concerned recruitment of the volunteers for the C cohort, which were both collaborators from the clinical center and people from the general public coming from the outside and thus not exposed to the hospital environment for long periods. As such, across these two subcohorts, no differences were found throughout their profiles, ruling out the hospital environment as a confounder factor.

A survey about easiness of execution of the breath collection test indicated that almost all volunteers (99%, Appendix A) from both groups defined the test as being “easy” or “very easy” to perform. This revealed the broad acceptance of the test as convenient by most of the volunteers.

## 5. Clinical Application and Future Outlook

This work reports the potential for using AI-based ML techniques on exhaled breath profiles as classifiers for detecting lung cancer in a set of LC patients vs. a healthy population, while discarding the qualitative information about specific (bio)chemical VOCs.

This multicentric study showed that no differences in the exhaled breath air profiles were found across volunteers recruited from both sites within the same group. A subsequent multicentric study has already been approved ethically for validating the method described here and will provide a proof of concept for this approach. This will confirm adequacy of both the patient preparation protocol and the classification algorithm under the overarching theme of standardization of an exhaled breath test for a clinical application.

Limitations must also be addressed. The first concerns the size of the study, which included a small group to demonstrate the broad usefulness of the test. This will be addressed in the subsequent validation study, which foresees the recruitment of a larger cohort. Second, environmental variability must be accounted for in a clinical or hospital setting. Although in our case, we did not find evidence of it, it may be relevant at other facilities or as a contributor to excessive modeling data and thus leading to results better than expected. Third, while this was a multicentric study, further population variability must be sought. As such, validation of the present procedure must be extended to additional clinical centers with different population characteristics. Fourth, strong efforts were put into standardizing operations—volunteer preparation, maintenance of equipment, and sample handling and management—meaning that there are several critical points requiring tight control.

Focusing on subgroup analysis, this work demonstrated that distinguishing beyond “*C* vs. *LC all stages*” groups was possible. In this way, screening for “*C* vs. *LC stage IA*” subgroup achieved 80% sensitivity and 93% specificity (90% accuracy). However, the most striking feature was the NPV (96%), confirming that this test is adequate for ruling out disease. This analysis was particularly important since we were considering samples from patients with early-stage pathology. Success in the classification task would allow us to conclude whether it would be possible to use VOCs present in exhaled air samples to diagnose early cases of lung cancer. Clinically, the analysis of these samples is extremely important as it will help health professionals in the difficult task of detecting new cases of the disease. Since the samples refer to an early stage of the disease, it was expected that the change in VOC values would not be as dramatic when compared to more advanced stages, leading to some extra challenges for the classification task. Histological classification was also attempted for differentiating “*Adenocarcinoma* vs. *Other histology*” with 75% for both sensitivity and specificity (75% accuracy). Despite promising, a larger population sample is required to elaborate a more robust model.

Accounting for the data analysis and classification, it is important to state and make clear that some aspects leave margin for improvement. First, data imbalance could be better handled, especially for “*C* vs. *LC stage IA*”, on which the percentage of controls is significantly higher (79%) than lung cancer cases (21%). This contributed to the lower classification of cancer positive cases, given by the lower PPV values. Secondly, considering the data preprocessing, data were not normalized due to the narrow interval of values [−0.046, 3.22]. Thirdly, the rich and large dataset with more than 4000 features could benefit from dimensionality reduction methods, although this path was not followed as computational resources were not an issue. Finally, data augmentation techniques could also be used, especially to address data imbalance. 

Concerning data overfitting, the results suggest that using 10-fold cross-validation, allowed the AI algorithms to effectively learn the true patterns in the training data. Since up to 90% of the samples were correctly classified in the test dataset, which the algorithms had not seen before, it can be concluded that the models did not overfit data. This supports the reliability of the results obtained.

## 6. Conclusions

Summarizing, from the results described here, we demonstrated that it may be possible to establish breath analysis for LC screening even without the burden of assigning specific molecular entities as potential biomarkers, which often leads to frustration. Instead, by leveraging the analysis of the exhaled breath profiles with the power of AI-based methods for the classification of the whole exhaled breath profiles, without considering qualitative information about specific (bio)chemical components, this work demonstrated that breath analysis can be used effectively for detecting LC at early stages against a healthy population. 

## Figures and Tables

**Figure 1 cancers-17-01685-f001:**
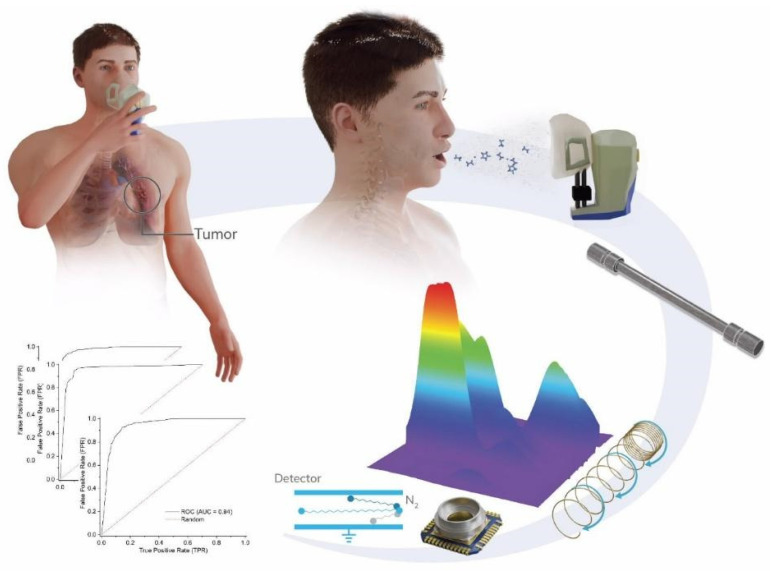
Schematic representation of workflow: exhaled breath sampling (top), analysis by GC-IMS (middle), and group classification (bottom).

**Figure 2 cancers-17-01685-f002:**
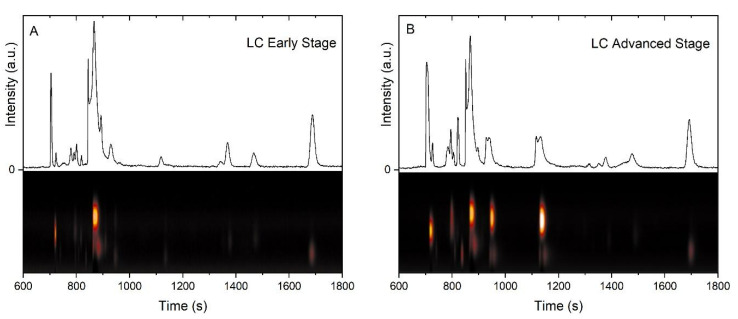
Two-dimensional chromatograms and their one-dimensional projections obtained by GC-FAIMS analysis for patients with lung cancer at early-stage (**A**) and late-stage (**B**), and a healthy patient (**C**). A superposition of all three profiles is also shown for comparison (**D**).

**Figure 3 cancers-17-01685-f003:**
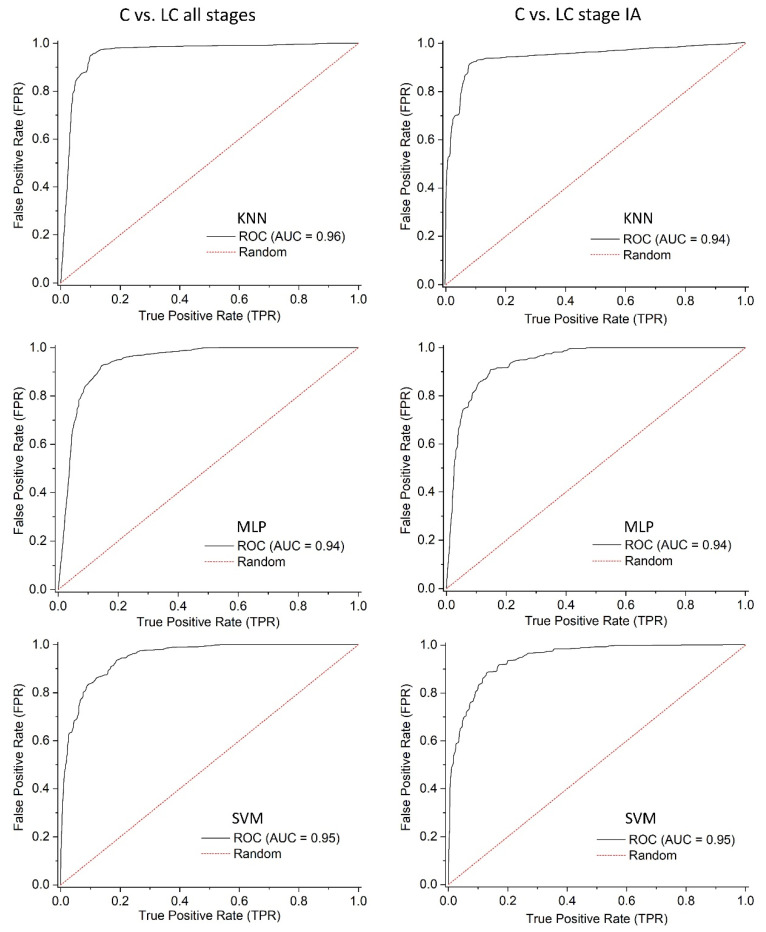
ROC curves for the classification models “*C* vs*. LC all stages*” and “*C* vs. *LC stage IA*” using the KNN, MLP, and SVM algorithms.

**Table 1 cancers-17-01685-t001:** Characterization of the study population systematized by group.

Parameter		Lung Cancer Patients	Control
*n* (%)		73 (37.2)	123 (62.8)
Age (years)			
	Median (range)	66 (41–86)	40 (20–78)
Sex, *n* (%)			
	Female	46 (63.0)	93 (75.6)
	Male	27 (37.0)	30 (24.4)
Smoking status, *n* (%)			
	Current smoker	23 (31.5)	39 (31.7)
	Former smoker	28 (38.4)	17 (13.8)
	Never smoker	22 (30.1)	67 (54.5)
Lung cancer type, *n* (%)			–
	Adenocarcinoma	49 (67.1)	
	Squamous cell	6 (8.2)	
	Small cell	5 (6.9)	
	Other	13 (17.8)	
Clinical stage, *n* (%)			–
	0	5 (6.8)	
	IA and IB	37 (50.7)	
	IIA and IIB	3 (4.1)	
	IIIA	10 (13.7)	
	IIIB and IIIC	3 (4.1)	
	IVA	11 (15.1)	
	IVB	4 (5.5)	

The dash “–“ means "not applicable".

**Table 2 cancers-17-01685-t002:** Results of the classification model for cohort segmentation of the algorithms tested.

Parameter/Method	SVM	MLP	KNN
*C* vs. *LC all stages*			
Sensitivity	88%	84%	87%
Specificity	86%	90%	92%
Positive Predictive Value (PPV)	77%	85%	88%
Negative Predictive Value (NPV)	93%	90%	92%
Accuracy	87%	88%	90%
*C* vs. *LC stage IA*			
Sensitivity	87%	75%	80%
Specificity	90%	94%	93%
Positive Predictive Value (PPV)	57%	77%	70%
Negative Predictive Value (NPV)	98%	93%	96%
Accuracy	89%	90%	90%
*Adenocarcinoma* vs. *Other Histology*			
Sensitivity	71%	77%	75%
Specificity	72%	78%	75%
Positive Predictive Value (PPV)	77%	81%	79%
Negative Predictive Value (NPV)	65%	73%	72%
Accuracy	71%	77%	75%

**Table 3 cancers-17-01685-t003:** AUC obtained for the best algorithms with 95% confidence intervals.

Method	AUC (“*C* vs. *LC All Stages*”)	AUC (“*C* vs. *LC Stage IA*”)
KNN	0.9486 (0.9327–0.9636)	0.9253 (0.8950–0.9532)
MLP	0.9331 (0.9163–0.9493)	0.9325 (0.9069–0.9547)
SVM	0.9428 (0.9274–0.9565)	0.9428(0.9229–0.9603)

**Table 4 cancers-17-01685-t004:** Pairwise comparison between the best algorithms for DeLong’s test over AUC’s *p*-values.

Benchmarking	*p*-Value (“*C* vs. *LC All Stages*”)	*p*-Value (“*C* vs. *LC Stage IA*”)
KNN vs. MLP	0.9792	0.9880
KNN vs. SVM	0.9920	0.9713
MLP vs. SVM	0.9873	0.9832

## Data Availability

The data that made possible to conduct this study are available from the corresponding author on reasonable request and once approved by the Data Protection Officer from author’s affiliation institution.

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
