# Peer review of "Breath Insights: Advancing Lung Cancer Early-Stage Detection Through AI Algorithms in Non-Invasive VOC Profiling Trials"

_cancers, 2025, doi:10.3390/cancers17101685_

Round 1
Reviewer 1 Report
Comments and Suggestions for Authors
This manuscript undertakes an examination of the application of artificial intelligence in the analysis of volatile organic compound (VOC) profiles derived from exhaled breath, with the objective of facilitating early detection of lung cancer. It integrates gas chromatography-ion mobility spectrometry (GC-IMS) with supervised machine learning methodologies, including support vector machines (SVM), multilayer perceptron (MLP), and k-nearest neighbors (KNN), demonstrating notable performance across various stages of cancer as well as different histological classifications.
Some areas require additional details /or explanations to enhance the manuscript quality:
1. The manuscript can benefit from additional details about preprocessing steps that include normalization methods and dimensionality reduction techniques, and how the data imbalance between healthy and cancer groups was handled.
2. The paper could benefit from a clearer explanation of why KNN, SVM, and MLP were selected over Random Forest or XGBoost classifiers. It would be wonderful to see performance metrics for all models included, especially for KNN. Additionally, the manuscript might be strengthened by presenting information about confounding factors, such as age, sex, smoking history, and medication use. Were these matched or adjusted for? Breathomics studies require this information to be crucial.
4. Your results would gain more credibility if you added confidence intervals and statistical significance tests (e.g., DeLong’s test for ROC AUCs) to accuracy, sensitivity, and specificity measurements.
5. This paper should explain how GC-IMS differs in comparision to GC-MS while addressing the challenges in clinical interpretation.
The manuscript overall could benefit from a bit more clarity in its methodology and with enhanced discussions on both the technical aspects and clinical boundaries.
Author Response
Answers to Reviewer 1
This manuscript undertakes an examination of the application of artificial intelligence in the analysis of volatile organic compound (VOC) profiles derived from exhaled breath, with the objective of facilitating early detection of lung cancer. It integrates gas chromatography-ion mobility spectrometry (GC-IMS) with supervised machine learning methodologies, including support vector machines (SVM), multilayer perceptron (MLP), and k-nearest neighbors (KNN), demonstrating notable performance across various stages of cancer as well as different histological classifications.
Some areas require additional details /or explanations to enhance the manuscript quality:
A: Dear Reviewer, we truly appreciate the time you took to analyze the paper and the valuable comments on it.
Let us please make an initial remark. This study was made under an initial ethical protocol, which is closed and addresses only a dataset that is not as large as we would like. The study hereby described follows entirely the approved protocol and description of work. The reason for this article is that 1) the results are consistent and quite relevant, and 2) its publication is meant to reveal very strong evidences and serve as the basis for the next and larger validation study that is already planned and approved.
Thus, your comments are of the highest value and will be taking into account while elaborating the next publication. Please find our point-by-point answers below
- The manuscript can benefit from additional details about preprocessing steps that include normalization methods and dimensionality reduction techniques, and how the data imbalance between healthy and cancer groups was handled.
A: The preprocessing steps undertaken were already explained in the supplement. For instance, “having proceeded to concatenate both [datasets]” and “all columns with null values were eliminated”. Since the dataset values vary between [-0,046; 3,22], there is no need for further normalization. Surely, according to the good practices of ML would tell us to do that indeed. We can explicitly refer to this textually by justifying and expressing the dataset’s interval. Dimensionality reduction has been found not to be necessary in this case, despite the large number of features/dimensions. So, we preferred not to degrade the dataset’s resolution and use all features for classification, since the computational resources to handle all datasets were not a problem. This information was added for clarity. Again, ML good practices are a good playbook, but some studies, prefer to go straight to the point. So, we can admit the dataset imbalance has been maintained throughout the study, justified by the fact that data stratification did not change the results. We added such explanations in the revised manuscript. Furthermore, in the revised ESI document we also made a statement about the preprocessing.
- The paper could benefit from a clearer explanation of why KNN, SVM, and MLP were selected over Random Forest or XGBoost classifiers. It would be wonderful to see performance metrics for all models included, especially for KNN. Additionally, the manuscript might be strengthened by presenting information about confounding factors, such as age, sex, smoking history, and medication use. Were these matched or adjusted for? Breathomics studies require this information to be crucial.
A: The explanation is on the supplement. KNN, SVM, and MLP had the best results overall, both for 'C vs LC stage IA’ and ‘C vs LC all stages’. The performance metrics can be found on the ESI material as well. Concerning confounding factors, further information was added in the Discussion section of the revised manuscript.
- Your results would gain more credibility if you added confidence intervals and statistical significance tests (e.g., DeLong’s test for ROC AUCs) to accuracy, sensitivity, and specificity measurements.
A: We agree that adding confidence intervals and significance testing, such as DeLong’s test, would strengthen the interpretation of our results. These data have now been included in the revised manuscript, especially CI and DeLong’s test to compare the AUCs.
- This paper should explain how GC-IMS differs in comparison to GC-MS while addressing the challenges in clinical interpretation.
A: In our opinion this has been addressed in the first sub-section of the Discussion section. Yet, we understand that the reviewer has its point, and we reinforced the text. Furthermore, although we do not own a GC-MS system it would be interesting to compare both the spectrometer’s performance and outcome. We thank the reviewer for the insightful comment.
- The manuscript overall could benefit from a bit more clarity in its methodology and with enhanced discussions on both the technical aspects and clinical boundaries.
A: We have addressed this in the revised version of the manuscript.
Reviewer 2 Report
Comments and Suggestions for Authors
The authors present a manuscript that uses GC-IMS to distinguish lung cancer patients from healthy controls. This is an interesting study but I have a few points.
Abstract -
Could specify this"The enrolled patients had LC diagnosed at different stages."
Could specify how breath was collected.
what is the justification to use healthy controls rather than a non cancerous lung disease group?
This is not completely clear, but I assume you mean you didn't try to name the individual VOCs? "AI methods ranked the sets of exhaled breath profiles across both groups through training and validation steps, while qualitative information was deliberately not taking part nor influencing the results."
I believe there are other studies that report promising results in the absence of qualitative data but it is good to try to understand the origins of the biomarkers. I think without very large validation studies there is always a question mark as to whether AI algorithms are overfitting data meaning it may not be generalisable to a larger population.
"Conclusions: Evaluation of the global exhaled breath profiles using AI algorithms enabled LC detection demonstrating that qualitative information may not be required, thus easing the frustration that many studies have experienced so far."
Introduction
This is generally well written and well referenced, but I feel that you could include a bit more detail about the current breath studies for diagnosing Lung cancer. Particularly the diagnostic potential of previous studies and the technologies used. Just a brief summary but it seems like currently there is more focus on the disease aspects and not the previous approaches.
If I'm honest Figure 1 didn't really help explain the procedure undertaken and I know that is a thermal desoprion tube etc. So maybe some annotation may be beneficial?
Methods
What is the absorbent material in the thermal desorption tubes?
You don't appear to have instrument parameters for ther IMS or the GC parameters for the separation step?
In classification methods you are talking about VOCs but are these features actually ion fragments, obviously they are associated with specific VOCs but as the number of features used is usually much greater as multiple ions can be generated from a single VOC then it is important to make a distinction.
Results - the controls don't appear to be age matched with a significant difference in Median age and a much larger age range. What effects will this likely have on the data?
Needs better explanation ", a complete profile based on intensities"
could you describe these differences in terms of retnetion times and peak intensity? "As shown in Figure 2, the profiles from both groups differed in their overall shape 206
and total integrated intensity (area under the curve, AUC), with AUCs higher for ad- 207
vanced-stage patients (Figure 2B). The profile of the healthy individuals displayed the 208
lowest AUC of the set (Figure 2C). Figure 2D shows a superposition of all three profiles 209
stressing the differences among them."
Possibly need more information on training data vs. validation data. What happened to multiple samples from each patient - how were these used for training/validation?
Discussion - good comprehensive coverage of the data. Addressed limitations and future work. Would maybe like to see the authors go into more depth about the design of any larger study to validate these pilot results.
Author Response
Answes to Reviewer 2
The authors present a manuscript that uses GC-IMS to distinguish lung cancer patients from healthy controls. This is an interesting study but I have a few points.
A: Dear Reviewer, we truly appreciate the time you took to analyze the paper and the valuable comments on it.
Let us please make an initial remark. This study was made under an initial ethical protocol, which is closed and addresses only a dataset that is not as large as we would like. The study hereby described follows entirely the approved protocol and description of work. The reason for this article is that 1) the results are consistent and quite relevant, and 2) its publication is meant to reveal very strong evidences and serve as the basis for the next and larger validation study that is already planned and approved.
Thus, your comments are of the highest value and will be taking into account while elaborating the next publication. Please find our point-by-point answers below
A – Abstract
- Could specify this"The enrolled patients had LC diagnosed at different stages."
- We meant that patients with LC had disease at different stages. We have clarified this.
- Could specify how breath was collected.
A: We understand the reviewer’s concern. However, given that the Abstract is a filed limited to 250 words it is already at the maximum we cannot go into more detail here.
- What is the justification to use healthy controls rather than a non cancerous lung disease group?
A: We did not understand this comment. The healthy controls were individuals with no neoplasic disease. That does not mean that respiratory diseases such as asthma or DPOC were present. Again given the limited size of the Abstract we cannot change it.
- This is not completely clear, but I assume you mean you didn't try to name the individual VOCs? "AI methods ranked the sets of exhaled breath profiles across both groups through training and validation steps, while qualitative information was deliberately not taking part nor influencing the results."
A: IMS does not provide a way to ID the individual VOCs. Unlike mass spectrometry, databases are unavailable and working with standards would be like finding needles in a haystack. Hence, that is the reason why working with the full chromatographic profiles was the strategy chosen.
- I believe there are other studies that report promising results in the absence of qualitative data but it is good to try to understand the origins of the biomarkers. I think without very large validation studies there is always a question mark as to whether AI algorithms are overfitting data meaning it may not be generalisable to a larger population.
A: We agree with the view obviously as this is an issue common across (almost) all studies using limited amount of data. This is addressed in the revised manuscript.
- "Conclusions: Evaluation of the global exhaled breath profiles using AI algorithms enabled LC detection demonstrating that qualitative information may not be required, thus easing the frustration that many studies have experienced so far."
A: The objective of this sentence is to mention that one does not need to actually know anything about the actual specific qualitative chemical content. The overall chemical content may be taken as the separated mixture without any further information and then looking at key differences in the profiles from the different groups. The rationale is a clinical one rather than a chemical one, as the physicians under real-world application need an answer to the key question which is “does this individual has cancer?”. And this can be answered without knowing what’s inside specifically and therefore avoiding inconsistencies found in many previous studies. Yet we thank the reviewer for raising the point. In addition being this the Abstract with size limitations we will not change it.
B – Introduction
- This is generally well written and well referenced, but I feel that you could include a bit more detail about the current breath studies for diagnosing Lung cancer. Particularly the diagnostic potential of previous studies and the technologies used. Just a brief summary but it seems like currently there is more focus on the disease aspects and not the previous approaches.
A: We thank the reviewer for the comment. Indeed the information was already present in the original (refs 12-18) we have now and some text to draw attention to this point in the revised manuscript.
- If I'm honest Figure 1 didn't really help explain the procedure undertaken and I know that is a thermal desorption tube etc. So maybe some annotation may be beneficial?
A: Figure 1 is just that, a schematic representation. Yet, we revised the caption to make it clearer.
C – Methods
- What is the absorbent material in the thermal desorption tubes?
A: We have included this information in the revised manuscript
- You don't appear to have instrument parameters for ther IMS or the GC parameters for the separation step?
A: We have included this information in the revised manuscript
- In classification methods you are talking about VOCs but are these features actually ion fragments, obviously they are associated with specific VOCs but as the number of features used is usually much greater as multiple ions can be generated from a single VOC then it is important to make a distinction.
A: We thank the reviewer for pointing out this. IMS works with a very mild ionization technique (milder than the mildest MS ionization technique. We have added this in the Discussion section.
D – Results
- The controls don't appear to be age matched with a significant difference in Median age and a much larger age range. What effects will this likely have on the data?
A: We have addressed this issue in the Discussion section.
- Needs better explanation ", a complete profile based on intensities"
A: We have revised this passage and made it clearer.
- Could you describe these differences in terms of retention times and peak intensity? "As shown in Figure 2, the profiles from both groups differed in their overall shape and total integrated intensity (area under the curve, AUC), with AUCs higher for advanced-stage patients (Figure 2B). The profile of the healthy individuals displayed the lowest AUC of the set (Figure 2C). Figure 2D shows a superposition of all three profiles stressing the differences among them."
A: We have revised this as per the reviewer’s suggestion.
- Possibly need more information on training data vs. validation data. What happened to multiple samples from each patient - how were these used for training/validation?
A: We thank the reviewer for raising this question. This is explained in the revised ESI document annexed (page 5 and Table S1).
E – Discussion
- Good comprehensive coverage of the data. Addressed limitations and future work. Would maybe like to see the authors go into more depth about the design of any larger study to validate these pilot results.
A: This was already mentioned in the original manuscript, concerning a subsequent validation study already approved by the institutional review board.